# Use of a glycomics array to establish the anti-carbohydrate antibody repertoire in type 1 diabetes

Paul M. H. Tran[1,2], Fran Dong[3], Eileen Kim [1], Katherine P. Richardson[1], Lynn K. H. Tran[1], Kathleen Waugh[3], Diane Hopkins[1], Richard D. Cummings [4], Peng George Wang [5], Marian J. Rewers[3], Jin-Xiong She[1] & Sharad Purohit [1,6,7] ✉

Type 1 diabetes (T1D) is an autoimmune disease, characterized by the presence of autoantibodies to protein and non-protein antigens. Here we report the identification of specific anti-carbohydrate antibodies (ACAs) that are associated with pathogenesis and progression to T1D. We compare circulatory levels of ACAs against 202 glycans in a cross-sectional cohort of T1D patients (n = 278) and healthy controls (n = 298), as well as in a longitudinal cohort (n = 112). We identify 11 clusters of ACAs associated with glycan function class. Clusters enriched for aminoglycosides, blood group A and B antigens, glycolipids, ganglio-series, and O-linked glycans are associated with progression to T1D. ACAs against gentamicin and its related structures, G418 and sisomicin, are also associated with islet autoimmunity. ACAs improve discrimination of T1D status of individuals over a model with only clinical variables and are potential biomarkers for T1D.

The sugars within complex carbohydrates or glycans, are components of glycoproteins and glycolipids on the surface of all cell types including those of all microbes. Both bound and free glycans also originate from microbial flora, dietary sources, as well as normal metabolism of host glycoconjugates. Glyco-chemistry, encouraged by the diversity of the tumor glycome, has fueled research in identifying the immunological functions of these glycans in diseases such as cancers and autoimmune diseases[1–4]. Tumor cells have been found to incorporate non-human derived glycan modifications, such as N-glycolylneuraminic acid (NeuGc) onto glycoproteins[5]. Recognition of abnormal glycosylation in human diseases has raised interest in research for carbohydrate-based biomarkers and treatment of human diseases[6–8]. Using glycan microarrays, researchers have identified the presence of naturally occurring anti-carbohydrate antibodies (ACAs) in normal human serum[9–11]. Specific types of ACAs have been reported in human pathologic conditions such as cancer and auto-immune diseases[7,8,12]. The serum repertoire of ACAs to natural glycans are involved in the progression and pathogenesis of cancer and auto-immune diseases and have prognostic and diagnostic value for disease course and treatment outcome[3,12,13]. Presence of ACAs has been reported in rheumatoid arthritis[1], inflammatory bowel diseases like Crohn's disease[14,15], and multiple sclerosis[16]. An important feature of these anti-glycan antibodies is their levels persist during the life of an individual[17] suggesting that these could serve as surrogate markers for T1D.

Type 1 diabetes (T1D) is characterized by immune-mediated destruction of the insulin-secreting β-cells of the pancreas. T1D comprises the majority of cases of diabetes seen in childhood, accounts for

[1]Center for Biotechnology and Genomic Medicine, Medical College of Georgia, Augusta University, 1120 15th Street, Augusta, GA 30912, USA. [2]Department of Internal Medicine, Yale School of Medicine, New Haven, CT CT06510, USA. [3]Barbara Davis Center for Diabetes, University of Colorado Denver, Mail Stop A-140, 1775 Aurora Court, Aurora, CO 80045, USA. [4]Department of Surgery, Beth Israel Deaconess Medical Center, Harvard Medical School, Boston, MA 02115, USA. [5]School of Medicine, Southern University of Science and Technology, Shenzhen 518055 Guangdong, China. [6]Department of Obstetrics and Gynecology, Medical College of Georgia, Augusta University, 1120 15th Street, Augusta, GA 30912, USA. [7]Department of Undergraduate Health Professionals, College of Allied Health Sciences Augusta University, 1120 15th Street, Augusta, GA 30912, USA. ✉e-mail: spurohit@augusta.edu

~5–10% of all cases of diabetes mellitus in the USA, and perhaps accounts for an even higher percentage in those nations with lower rates of obesity[18]. A characteristic feature of T1D is the presence of auto-antibodies against antigens of pancreatic and non-pancreatic origin[19,20]. HLA genotyping and auto-antibodies to glutamate decarboxylase (GADA), islet cells (ICA), protein tyrosine phosphatase (IA-2A) and zinc transporter (ZnT8A) are used for diagnosis, prediction, and risk analysis of T1D[21–23], and additional antigens are being discovered[24]. These auto-antibodies have been extensively utilized for identification and risk stratification in both T1D patients and their first-degree relatives[25–28]. Auto-antibodies to ganglioside GT3 and the glycolipid sulfatide have also been identified in T1D patients[29].

Past efforts to identify environmental triggers of islet autoimmunity have used regular follow-up to collect information on diet, viral, and bacterial infections, among other factors[30–32]. We reasoned those environmental agents which can trigger islet autoimmunity will result in generation of ACAs and may be identified through serologic detection of immunoglobulins recognizing carbohydrates in a high-throughput multiplex assay. This approach allows identification of exposures not collected in current cohort studies and can also identify exposures occurring during periods between the regular follow ups.

Here we show that serum levels of ACAs correspond with functional glycan classes and that certain classes are strongly associated with islet autoimmunity and progression to T1D in high-risk individuals recruited in the Diabetes Autoimmunity Study in the Young (DAISY). Notably, ACAs targeting *Micromonospora* spp derived aminoglycosides were associated with progression to T1D, while ACAs targeting the *Streptomyces* spp derived aminoglycosides were not. This suggests an influence for environmental exposures in the differential prevalence of aminoglycoside targeting ACAs and risk of T1D progression. Finally, we show that a prediction model with both clinical variables and ACAs clusters is better able to discriminate T1D progressors vs non-progressors as compared to a clinical variable only model or a combined random forest model with both clinical variables and ACA.

## Results

### Diverse ACAs in T1D patients and controls

Serum ACAs were profiled against 202 naturally occurring glycans using a bead-based suspension array developed in-house[33] (Fig. 1a). Details on the validation of the assay have been previously published, but we re-assessed the analytical parameters of the assay for select glycans in this study (Supplementary Figs. 1–6). Based on their structure and functional roles in cellular localization and recognition, these glycans were divided into 22 major functional classes, including glycolipid antigens, blood group antigens, Lewis antigens, monosaccharides, chitin derivatives and many glycans that are expressed in algae and plants. A list of the glycans, their structures, and functional classes are provided in Supplementary data. The participants profiled for ACAs were from the Phenome and Genome of Diabetic Autoimmunity study (PAGODA, Georgia, USA) and Diabetes Autoimmunity Study in the Young (DAISY, Denver, Colorado, USA). The ACA levels were profiled in 576 participants from the PAGODA study, which is a case-control study with limited clinical data (Table 1). We measured ACA levels in the earliest serum sample available from 112 participants from the DAISY study, which is a cohort study with monitored from birth to development of T1D and contains an extensive clinically relevant data for progression to T1D (Table 2).

As found in previous studies[29,34], varying levels of ACAs were detected in both the PAGODA and DAISY cohorts against several classes of glycans. Antibodies to monosaccharides (#5), blood group and Lewis antigens (#35, 36, 60, 128, 129,), isoglobo series (#155, 156), foreign antigens viz., αGal antigens (#149,152), Forssmann antigens (#183,184) and aminoglycosides (#96-102) were detected at the highest abundance (Fig. 1b and Supplementary Table 1). This is in general agreement with previous ACA profiling studies[35,36]. Some individuals in the DAISY longitudinal study had multiple serum samples available from different time points, and we were able to see stability of ACA detection over time (Supplementary Fig. 7).

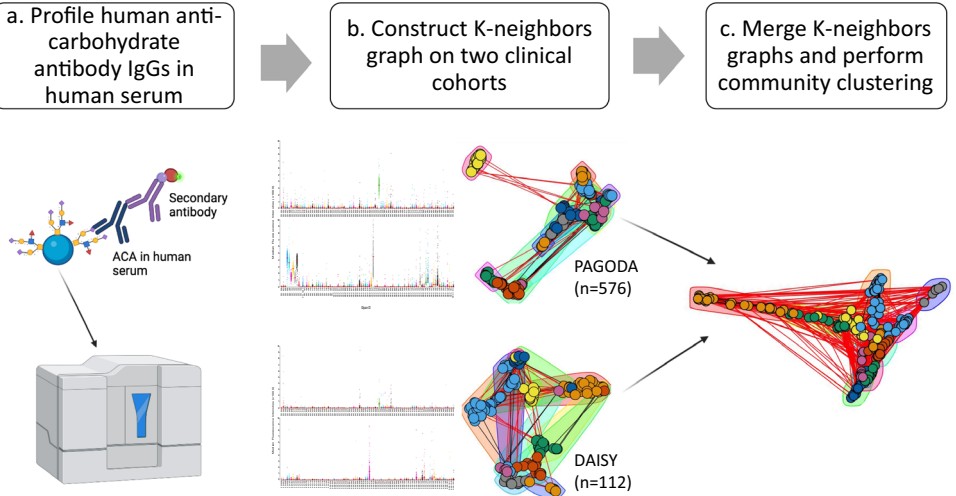

Fig. 1 | Community clustering of 201 glycans in serum samples from 688 participants identify 11 major clusters. a Overview of multiplex glycan suspension array platform and methodology. Briefly, diluted human serum is added to a mixture of fluorescent beads each linked to a different glycan structure to allow human anti-carbohydrated antibodies (ACAs) to bind to the immobilized glycans. A biotin-conjugated anti-human IgG antibody is added. Finally, streptavidin-conjugated phycoerythrin is added and the median fluorescence intensity is measured on a Luminex instrument. Created with BioRender.com. b Distribution of ACAs in the Diabetes Autoimmunity Study (DAISY) and Phenome and Genome of Diabetic Autoimmunity (PAGODA) shown as median fluorescence intensity dot plots. 5-nearest neighbor graphs were constructed based on cosine similarity and projected onto 2 dimensions using the UMAP algorithm for ACAs detected. Each color is arbitrarily chosen to represent a different cluster using the Louvain algorithm. c The PAGODA and DAISY k-nearest neighbors graphs were merged together. Edge weights unique to each graph were kept as is. For edges where both the PAGODA and DAISY graphs had weights, the product was subtracted from the sum of the two weights. The Louvain clustering algorithm was applied to identify glycan communities (arbitrary color for each cluster). Source data are provided as a Source Data file.

**Table 1 | Clinical characteristics of participants of Phenome and Genome of Diabetes Autoimmunity (PAGODA) Study**

| Characteristic | Control, $N = 272$ | Non-progressor, $N = 26$ | Progressor, $N = 278$ | p value[b] |
|---|---|---|---|---|
| Sex | | | | 0.9 |
| Female, n (%) | 146 (54%) | 13 (50%) | 153 (55%) | |
| Male, n (%) | 126 (46%) | 13 (50%) | 125 (45%) | |
| FDR, n (%) | 149 (55%) | 18 (69%) | 49 (18%) | <0.001 |
| Draw Age[a] | 18 (7, 41) | 8 (5, 17) | 21 (14, 42) | <0.001 |
| HLA_risk, n (%) | | | | <0.001 |
| H | 35 (14%) | 9 (35%) | 83 (39%) | |
| L | 216 (86%) | 17 (65%) | 129 (61%) | |
| Unknown | 21 | 0 | 66 | |

All p values were two-sided and a $p < 0.05$ was considered significant.
[a]Median (IQR).
[b]Pearson's Chi-squared test; Kruskal–Wallis rank sum test.

**Table 2 | Clinical characteristics of participants of Diabetes Autoimmunity Study in the Young (DAISY) Study**

| Characteristic | Control, $N = 30$ | Progressor, $N = 47$ | Non-progressor, $N = 35$ | p value[b] |
|---|---|---|---|---|
| Sex, n (%) | | | | 0.8 |
| Female | 17 (57%) | 24 (51%) | 17 (49%) | |
| Male | 13 (43%) | 23 (49%) | 18 (51%) | |
| FDR, n (%) | | | | 0.5 |
| No | 15 (50%) | 17 (36%) | 15 (43%) | |
| Yes | 15 (50%) | 30 (64%) | 20 (57%) | |
| Draw Age[a] | 8.0 (5.4, 11.1) | 6.6 (4.1, 8.5) | 8.9 (4.1, 10.6) | 0.1 |
| HLA risk | | | | 0.4 |
| H | 18 (60%) | 35 (74%) | 25 (71%) | |
| L | 12 (40%) | 12 (26%) | 10 (29%) | |

All p values were two-sided and a $p < 0.05$ was considered significant.
[a]Median (IQR).
[b]Pearson's Chi-squared test; Kruskal–Wallis rank sum test.

## ACA networks enriched for functional classes

The type and structure of glycans have an important function in glycoprotein structure and function inside the cells[37], as well as in control of innate and adaptive immune responses[38,39]. Since many of the glycan structures we assessed had overlapping substructures and linkages which can be recognized by the same ACA, the ACA levels against these glycans have a degree of interdependence and cannot be individually analyzed. The ACA data from both clinical cohorts were individually subjected to community clustering[40] to account for ACAs which recognize similar idiotypes (Fig. 1b). We then combined all the individuals from each cohort to create a merged k-nearest neighbors (KNN) graph to identify closely related ACAs. Community clustering of the merged KNN graph led to identification of 11 major ACA clusters (Fig. 1c). These clusters are mostly unique from each other with some correlation observed between Clusters 1 and 8 and among Clusters 2, 3, 5, and 10 (Supplementary Fig. 8).

The ACA clusters were found to be enriched for functional glycan classes (Fig. 2a). Cluster 1 was enriched for ACAs against mono-, di-, and tri-saccharides, and glycolipids. Cluster 2 was enriched for ACAs against blood group oligosaccharide analogues, chitin, ganglio-series analogues, globo-series and globo-series analogues, glucuronylated oligosaccharides, maltodextrins, and stage-specific embryonic antigens. Cluster 3 was enriched for ACAs against ganglio-series and O-linked glycans. ACAs against blood group H antigens were enriched in Cluster 4. ACAs against both blood group A and B antigens were enriched in Cluster 5. Cluster 6 is enriched for ACAs against lacto-series and plant oligosaccharides. Cluster 7 and 9 were not enriched for ACAs against any specific glycan class. Cluster 8 is enriched for ACAs against aminoglycoside antibiotics and glycolipids. Cluster 10 is enriched for ACAs against maltodextrins. Cluster 11 is enriched for ACAs against isoglobo-series and Lewis antigens.

## T1D associated with ACA clusters

We assessed ACA cluster association with T1D phenotypes in the DAISY cohort since it is more clinically balanced than the PAGODA cohort (Tables 1 and 2). All glycans belonging to individual ACA clusters were subjected to principal component analysis (PCA), and the first PC values from each cluster were then subjected to linear regression. Our linear regression models show that the first principal components of ACA clusters are associated with the development of auto-antibody and progression to T1D (Fig. 2b). Higher levels of ACAs against glycans in cluster 7 were associated with increased age at blood draw. No ACA clusters were associated with first degree relative with T1D, HLA risk, or sex. Higher levels of ACAs in clusters 2, 5, and 7 are associated with progression to T1D, whereas lower levels of ACAs in clusters 1 and 8 (enriched for aminoglycosides) are associated with progression to T1D. Higher levels of ACAs against glycans in cluster 3 (enriched for ganglio-series and O-linked glycans) were associated with islet auto-immunity and T1D.

## Oligosaccharides associated with T1D progression

We performed univariate analysis on the glycan classes enriched in the islet autoimmunity and T1D associated clusters to discern substructure patterns. ACAs against the monosaccharides βGalNAc (#7) and αmannose (#16) were associated with progression to T1D (Supplementary Fig. 9). These are both natural monosaccharides expressed in the human body[41]. In contrast, ACAs against foreign monosaccharides, such as αrhamnose (#5), and D-rhamnose (#72) were not associated with T1D. Additionally, the ACAs associated with progression are not the most abundant ACAs in circulation, which are αrhamnose (#5), αGlcNAc (#21), and βGlcNAc (#6)[13,35].

ACAs against the disaccharides Galβ1,4Glcβ (#15) and Galβ1,4-GalNAcβ (#17) were associated with progression to T1D (Supplementary Figure 9). In contrast, ACAs against the disaccharides Galβ1,3GlcNAcβ (#18) and Galβ1,3GalNAcα (#27) were not associated with islet autoimmunity or progression. This suggests that the β1,4 Gal linkage may be associated with T1D progression compared to the β1,3 Gal linkage.

ACAs against the trisaccharides Galα1,3Gal-β1,4Glcβ (#21) and GlcNAcβ1,3Galβ1,4Glcβ (#28) were associated with progression to T1D (Supplementary Fig. 9). However, there are no consistent structural differences between these two glycans and the remaining four glycans in the trisaccharide group.

## Blood group antigens associated with islet autoimmunity and T1D

Of the five blood group A antigens assessed, ACAs against two glycans (#125,127) were associated with both islet autoimmunity and T1D progression (Fig. 3). Of the five blood group B antigens assessed, only ACAs against #131 were associated with islet autoimmunity and T1D progression.

## Glycolipids associated with T1D progression

ACAs against the glycolipids Neu5Acα2,3Galβ1,3GlcNAcβ (#22), Neu5Gcα2,6Galβ 1,3GlcNAcβ (#25), and Neu5Gcα2,6,Galβ1,4Glcβ (#31) were associated with T1D progression. ACAs against the six remaining glycolipids were not associated with islet autoimmunity or T1D progression (Fig. 3).

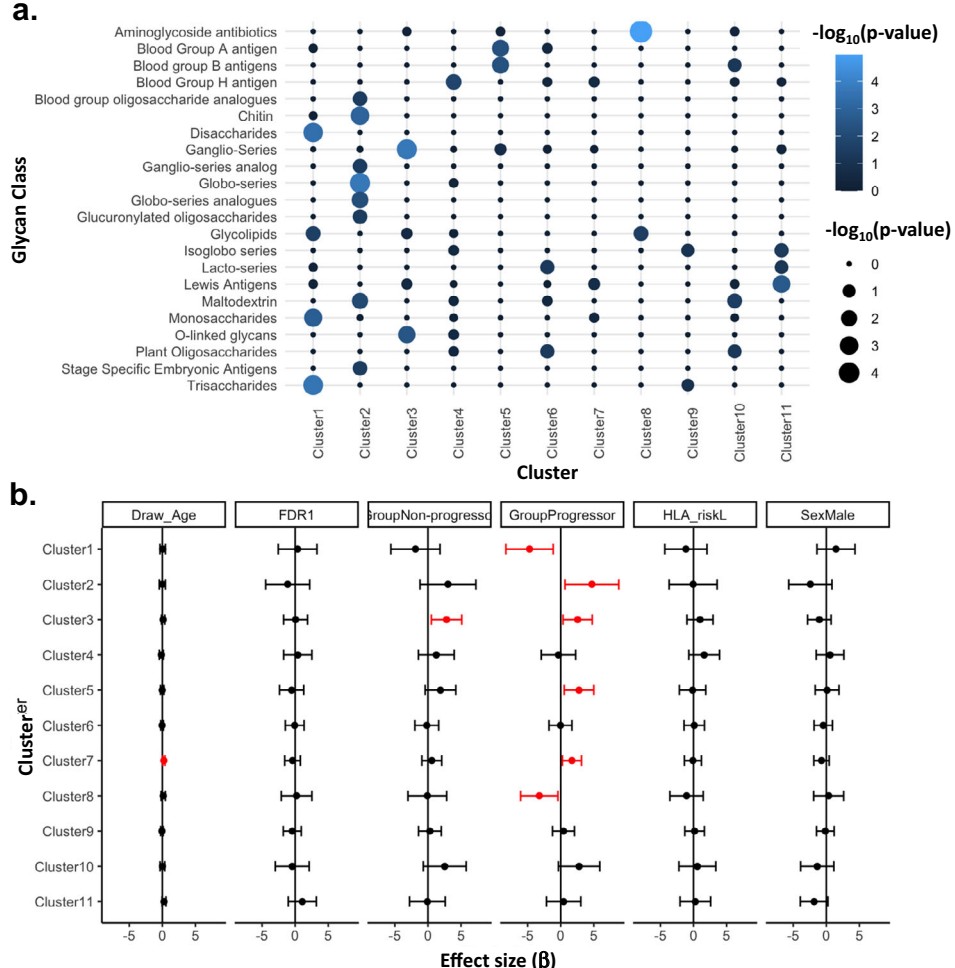

**Fig. 2 | ACA community clusters are enriched for functional glycan classes and associated with type 1 diabetes phenotypes. a** Bubble chart showing glycan classes enriched (lighter blue and larger bubble size) in each ACA cluster calculated using fisher's exact test. **b** Forest plot showing linear model for the first principal component of each glycan cluster against the clinical variables age at blood draw, presence of first degree relative with T1D, non-progression to T1D, progression to T1D, HLA risk, and sex in DAISY participants (*n* = 112). Effect size (dots) and 95% confidence interval (bars) are presented. All *p* values were two sided and a *p* < 0.05 was considered significant (significant associations are displayed as red, while non-significant associations are displayed as black). Source data are provided as a Source Data file.

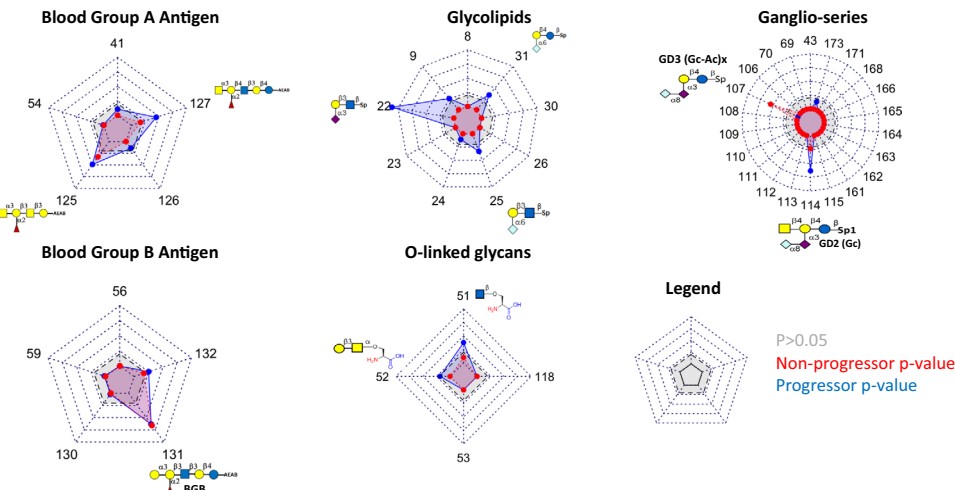

**Fig. 3 | ACAs differently detected in non-progressors and progressors compared to controls.** Radar charts showing −log10(*p* value) with blue representing association with progression and red representing association with non-progressors. The gray shadowed area represents non-significances (*p* > 0.05). Shaded area in red shows significant associations with non-progressors status (red dots and shaded area). Blue dots and shaded area shows significant associations with progression to type 1 diabetes (progressors). BGB: blood group B antigen, GD3: ganglioside GD3, GD2: ganglioside GD2, Gc-Ac: N-Acetyl-Glycolylneuraminic acid, Gc: N-Glycolylneuraminic acid. Source data are provided as a Source Data file.

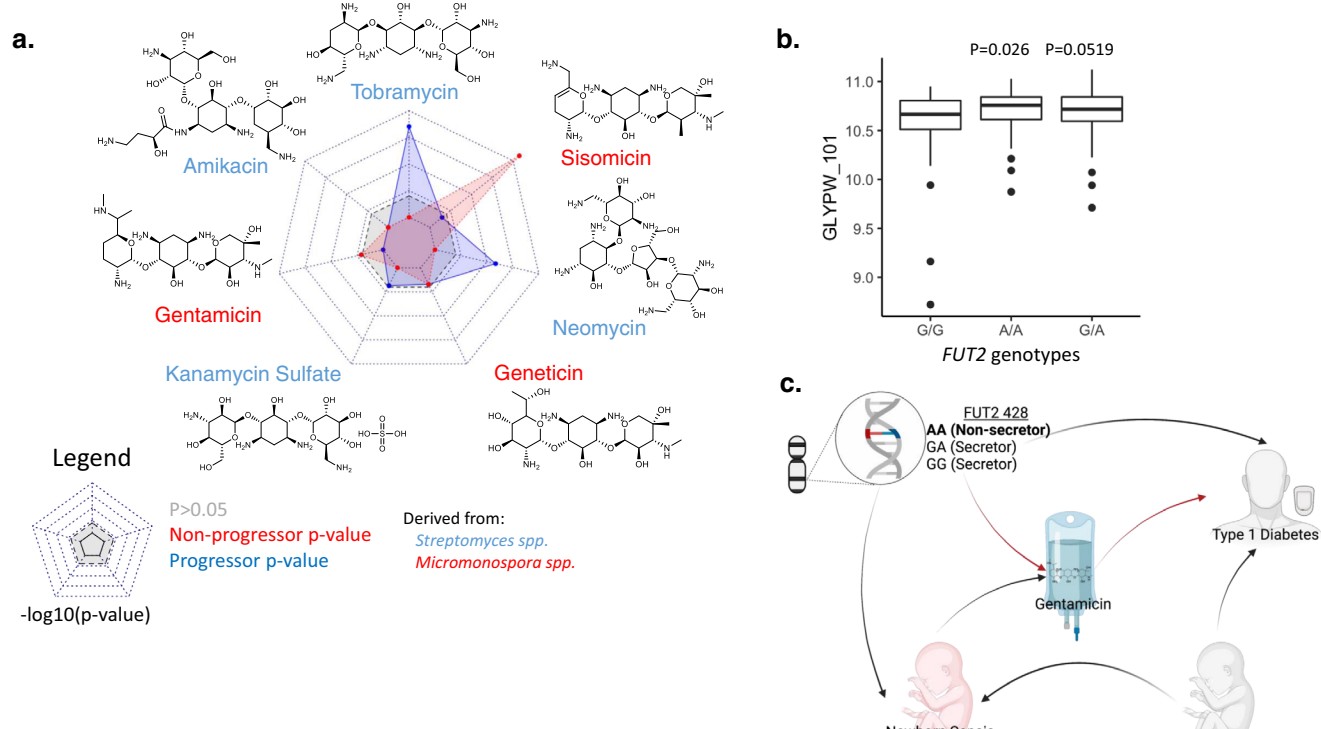

**Fig. 4 | ACAs differently detected in non-progressors and progressors compared to controls. a** Radar charts showing −log10(*p* value) with blue representing association with progression (blue dots and shaded area) and red representing association with non-progressors (red dots and shaded area). The gray shadowed area represents non-significances (*p* > 0.05). Aminoglycoside structures drawn using ChemDraw version 19.1.0.8. **b** Boxplot of ACA levels against gentamicin for fucosyltransferase-2 (*FUT2*) (rs601338) genotypes G/G (*N* = 41), A/A (*N* = 39), and G/ A (*N* = 84). Differences between the *FUT2* genotype groups were evaluated by Tukey's test accounting for multiple testing with Benjamini-Hochberg method,

horizontal lines on the box represent the 25th, 50th and 75th percentile, the whiskers represent maximum and minimum values of anti-gentamicin IgG values. All *p* values were two-sided and a *p* < 0.05 was considered significant. **c** Directed acyclic graph of report associations related to *FUT2* and T1D[51,52], *FUT2* and neonatal sepsis[48], prematurity and T1D[58], prematurity and neonatal sepsis[56], neonatal sepsis and gentamicin[56], and our reports of *FUT2* association with ACAs against gentamicin and of ACAs against gentamicin and T1D. Created with BioRender.com. spp.: species. Source data are provided as a Source Data file.

## Ganglio- and O-linked glycans in islet autoimmunity and T1D

ACAs against ganglioseries have been reported to be enriched in new-onset T1D patients[42,43]. We found ACA against one of four GD3 antigens (#107) associated with islet autoimmunity and one of two GD2 antigens (#114) associated with both islet autoimmunity and T1D progression (Fig. 3). ACAs against GM2 (#43, 69, 115) and GT3 antigens (#110, 111) were not associated with islet autoimmunity or T1D.

ACAs against the O-linked glycans GlcNAcβ-O-Ser (#51) were associated with islet autoimmunity and T1D progression while those against Galβ,3GalNAcα-O-Ser (Gal-Tn-Antigen, #52) were associated with T1D progression (Fig. 3).

## Aminoglycosides associated with islet autoimmunity

In our ACA data, higher levels of ACA against gentamicin (#98), geneticin (#100), and sisomicin (#102) were detected in individuals persistently positive for islet autoantibodies compared to controls (Fig. 4a). These three aminoglycosides are all derived from the *Micromonospora* spp[44,45] and all three contain the carbohydrate garosamine[46], suggesting that the ACAs may target garosamine. By comparison, the remaining four aminoglycosides (#96, #97, #99, #101) profiled are all derived from *Streptomyces* spp and lack the glycoside garosamine in their structure. The density distribution of the median fluorescence intensity for ACAs against gentamicin shows a bimodal distribution with 5.3% of individuals in the higher MFI peak (Supplementary Fig. 10). This is similar to the rate of gentamicin administration for suspected neonatal sepsis[47].

## *FUT2* genotype associated with anti-gentamicin IgG levels

Previous groups have shown a potential association between the *FUT2* genotype and neonatal sepsis risk[48]. In addition, the *FUT2* locus is a well-documented autoimmune disease associated region[49,50], including for T1D[51,52]. We assessed if the anti-aminoglycoside IgG level association with islet autoimmunity and T1D could be mediated by the *FUT2* genotype. *FUT2* non-secretors (A/A) have higher anti-gentamicin IgG levels than *FUT2* secretors (G/G) on average (Fig. 4b). We constructed a directed acyclic graph of the known associations between *FUT2*, neonatal sepsis, prematurity, and T1D, and included associations between anti-gentamicin IgG levels and *FUT2* and anti-gentamicin IgG levels and T1D (red, Fig. 4c).

## ACA improve discrimination of T1D status

We assessed if ACAs can improve discrimination of T1D status in the DAISY cohort using a binomial model with progressors and controls. The full model with all ACA clusters and clinical variables improved discrimination of T1D and control compared to a null model with only clinical variables (LRT *p* = 0.0056, Fig. 5a). Based on permutation testing, the full model was a robust predictor of T1D status (*q* = 0.0291) compared to the 8 ACA cluster model (*q* = 0.2084) and the clinical variable only model (*q* = 0.9878). The glycan clusters account for 18.4% of variance of T1D status while all genetic components together, including sex, first degree relative with T1D status, and HLA risk account for 9.8% of variance in the DAISY cohort (Fig. 5b). We assessed if enough ACAs have been profiled for maximal discrimination of T1D status by performing bootstrapping without resampling with

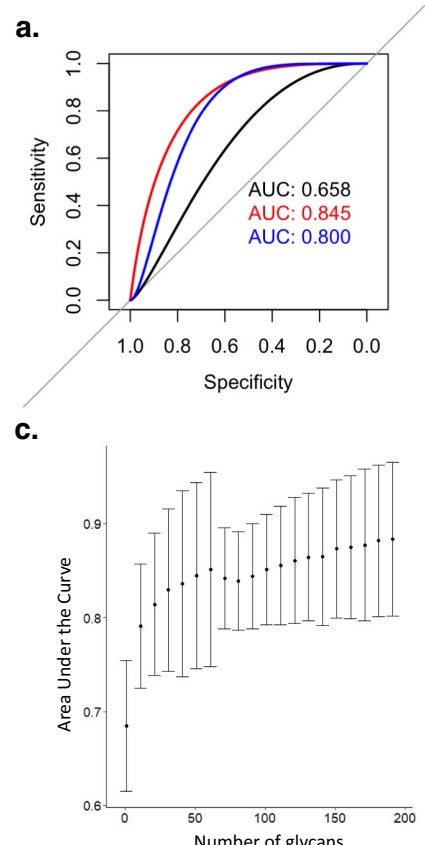

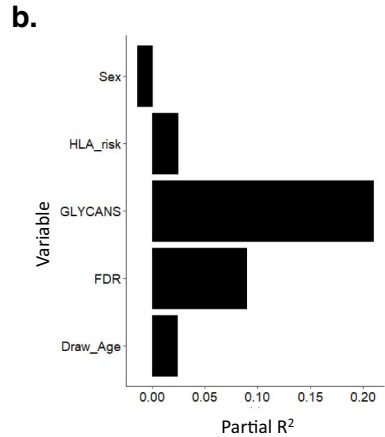

**Fig. 5 | ACAs improve prediction of T1D. a** Receiver-operator characteristic curve for three predictive models of T1D. A null model including sex, HLA risk, FDR, and draw age (black). A full model including all variables in the null model plus the first principal component from each of the 11 clusters of ACA (red). A partial model including all variables in the null model plus the first principal component from each of the 5 clusters of ACA significantly associated with progressors or non-progressors (blue). **b** The variance accounted for by each component of the full model from A. Partial correlation coefficient ($R^2$) values from full model was plotted (black bars) against the variables (x-axis). **c** The area under the receiver-operator characteristic curve for a random number of glycans included in a ridge regression model with clinical variables from 1 to 200. AUC (dots, y-axis) values were plotted against number of glycans (x-axis), the dots represent mean AUC value while error bars represent 95% confidence intervals for the AUC values. Source data are provided as a Source Data file.

increasing number of glycans and found that ACAs profiled against 50 glycans in the DAISY cohort was sufficient for T1D status discrimination with an AUC of 0.85 (Fig. 5c). We compared our discrimination analysis to random forest, a well-established machine learning method that can account for data with high correlation structure. We found that a random forest model of 202 ACA features to discriminate control and progressors in the DAISY cohort is not significantly different from the null distribution based on permutation analysis ($q = 0.063$).

## Discussion

We present a large-scale profiling study of ACAs against 202 glycans in two T1D cohorts totaling 688 samples and individuals (Fig. 1). Clusters enriched for mono-, di-, and tri- saccharides, aminoglycosides, blood group A and B antigens, and glycolipids were associated with progression to T1D. A recent T1D GWAS meta-analysis[51] shows a potential association between T1D and the ABO genetic locus, although these findings vary with the ABO prevalence of the cohort studied[40,53]. Other studies have shown potential associations between autoimmune disease and ABO blood type[40,54] albeit in smaller cohort sizes. Clusters enriched for ganglio-series and O-linked glycans were associated with both islet autoimmunity and progression to T1D (Fig. 3).

The *FUT2* genetic locus has been associated with T1D in T1D GWAS[51,52] and with multiple other autoimmune diseases[49,50]. The ABO, *FUT2* genetic, and ACA associations with T1D may be related to changes in the microbiome[55]. We showed the *FUT2* genotype is associated with ACAs against aminoglycosides (Fig. 4b).

Aminoglycoside exposure occurs for newborns suspected of neonatal sepsis[56]. The individuals are treated with gentamicin. Approximately 5–10% of the population is treated with gentamicin for suspected neonatal sepsis after birth[47], which matches the proportion of individuals with higher ACA against gentamicin (5.3%). In addition, premature infants are at higher risk of developing neonatal sepsis[57] and of developing type 1 diabetes[58] (Fig. 4C). Further research is required to discern the causal relationships between prematurity, altered microbiome, aminoglycoside exposure, and type 1 diabetes.

Currently, oligosaccharides (including mono-, di-, and tri-) and blood group antigens have not been associated with progression to T1D. Glycolipids and gangliosides have been associated with progression to T1D. We found that IgG antibodies against GM2 and GT3 were not associated with islet autoimmunity or T1D as previously reported[42,43] (Fig. 3, #69 and #110, respectively). Instead, our study identified the GD2 and GD3 gangliosides were associated with progression to T1D. The previous studies were limited by smaller sample sizes and less glycans to assess. Additionally, the previous studies used ELISA based assays whereas we applied a bead-based study. Thus, different epitopes could be exposed for these glycans. Different glycan densities can also affect antibody affinity and this can vary across assays and platforms[59].

While all four are gangliosides, GM2 is an a-series ganglioside, GD2 and GD3 are b-series gangliosides, GT3 is a c-series ganglioside. A-, b-, and c-series gangliosdies have 1, 2, and 3 sialic acid residues, respectively. Interestingly, a biochemical study of the glycolipids showed that

GM3 makes up the majority (66.7%) of total gangliosides detected in the whole pancreas whereas an unspecified GM2 co-migrating glycolipid makes up the majority (74.2%) of total gangliosides detected in pancreatic islets[60]. The co-migrating glycolipid is likely GD2 since the number of sialic acid residues have a strong influence in glycolipid migration in both HPLC and HPTLC. This data suggests that GD2 ganglioside is expressed more in pancreatic islets and that ACAs against GD2 are associated with progression to T1D.

We developed a cluster-based analysis method for ACA profiling since many of the glycan structures tested are similar and the signals from ACA to these different glycans are likely dependent. However, since the idiotype targeted by the ACA were not known beforehand, we used an unsupervised clustering approach rather than using the known functional glycan classes. The unsupervised clusters were enriched for these glycan classes (Fig. 2A), which further reassured us of this approach.

ACA profiling provides a method to identify environmental exposures associated with disease. It is advantageous over viral and microbial sequencing methods which only provide a snapshot at the time of sample collection. Instead, ACA profiling depends on the serologic memory of the environmental exposure.

The limitations of our study include a small cross-sectional study design with most serum samples collected after islet autoantibody seroconversion. In addition, only assays with known glycan binding proteins could be validated, while others were not. Hence, our findings require validation in a larger longitudinal cohort.

In summary, our data demonstrated that ACAs are associated with islet immunity and progression to T1D. The combination of ACA and cluster-based analysis accounts for more variance than clinical variables alone for the T1D phenotype (18.4% vs 9.8%). We found associations between *FUT2* genotype, ACA against gentamicin, and islet autoimmunity, which suggests the potential influence of environmental agents in T1D pathogenesis that requires validation in a larger, longitudinal cohort.

## Methods

### Blood collection from human participants

The study was performed according to Declaration of Helsinki (1997), and approved by the institutional review boards at Augusta University, Augusta, GA and University of Colorado, Denver, CO. Written informed consent was obtained from all participants or their parents. The samples used in the study were obtained from participants of the Phenome and Genome of Diabetes Auto-immunity (PAGODA)[61] and Diabetes Autoimmunity Study in the Young (DAISY) study[62]. PAGODA study is a cross-sectional design and contained controls ($n = 278$) and T1D patients ($n = 298$). 112 children at high risk for T1D from DAISY were followed for development of autoantibody or progression to T1D and they were divided into three comparison groups: progressors ($n = 47$), non-progressors ($n = 35$), and controls ($n = 30$). Progressors were children who progressed to type 1 diabetes (T1D). Non-progressors are children having at least two consecutive positive visits for islet autoantibodies by radiobinding assay (RBA), but did not develop T1D at their last follow-up. The control are individuals who are neither progressors nor non-progressors. The islet autoantibodies included insulin, GAD, IA-2 and ZnT8. The sex and age of the serum sample, as well as HLA-DQB1 genotypes for all participants were summarized in Tables 1 and 2. Serum samples were collected in Tiger top tubes and processed and stored at −80 °C within 45 min of collection. Repeated freezer/thawing cycles were avoided for all samples.

### Construction of glycan array

Access to large quantities of diverse natural glycans remains a challenge in the field of glycobiology and the glycans published in ours and other similar studies represent those simple enough to be synthesized de novo[63] or those able to be isolated from a natural source[64]. Thus, our set of 202 glycans represents the current capability of synthetic and purification chemistry in the field of glycobiology.

All the glycans with free amino-terminal were sourced from the Consortium for Functional Glycomics (CFG), the laboratory of Dr. Peng Wang at Georgia State University, and the laboratory of Dr. Richard D. Cummings at Harvard Medical School and the National Center for Functional Glycomics. Glycans with 3 or 4 carbon spacer[63], glycans with bifunctional fluorescent tag, 2-amino(N-aminoethyl) benzamide (AEAB), and labelled glycans[64] with a free amino group were conjugated to fluorescent dye encoded microspheres containing carboxyl (-COOH) groups (Luminex Corp., Austin, Tx, USA). Glycans (2.5 ug) containing free-amino group were covalently linked to the beads by one-step method using 1-Ethyl-3-[3-dimethylaminopropyl] carbodiimide hydrochloride (10 mg/ml) in deionized water. Conjugation was carried out for 2 h at room temperature with shaking. After conjugation, the beads were collected from unreacted glycans. All unreactive sites were blocked with human serum albumin (1%, w/v) prepared in phosphate buffer saline[33].

### Measurement of ACA

For measurement of anti-glycan IgG, human serum samples were diluted 500-fold in 1% BSA in PBS. Briefly, 1000 microspheres for each glycan were added to each well and incubated with 10 μL of diluted serum sample for 2 h. After 2 h, unbound reagents were removed by vacuum suction and the wells were washed with wash buffer (PBS containing 0.1% Tween-20). Detection was performed by adding biotinylated anti-human IgG (3 μg/mL in 1% HSA/PBS Southern Biotech, AL, USA) to each well and the plates were incubated for 1 h. After the incubation, the plates were washed and incubated with SAPE (3 μg/mL in wash buffer) for 30 min. The plates were washed and the beads resuspended in 60 μL of wash buffer. The median fluorescent intensity (MFI) was captured on FlexMAP 3D reader (Millipore, Billerica, MA, USA) using xPONENT4.3 (v4.3, Luminex Corp, Tx, USA) with the following instrument settings: events/bead: 50, minimum events: 0, flow rate: 60 μL/min, sample size: 50 μL and doublet discriminator gate: 8000-13500. Raw data files from Luminex FM3D machine were processed using a software pipeline, which we have developed for quality control, visualization and normalization of raw Luminex data[33]. No glycan beads (3 bead regions) were included within each well to control for background noise. The average MFI for no glycan beads were then subtracted from the glycan beads to determine anti-glycan antibody positivity.

### Fucosyltransferase 2 (G428A) polymorphism

Polymorphism rs601338, in *FUT2* gene, was determined in DNA samples from the participants of PAGODA study using Taqman probe Assay ID C__2405292_10 (Applied Biosystems). The reaction mixture consisted of 2× Genotyping master mix (cat#4371353, Applied Biosystems), 20 ng of DNA, and probes, the cycling conditions recommended by the manufacturer.

### Analyses performed for each T1D cohort

The samples analyzed from the PAGODA study were cross-sectional with a random selection of type 1 diabetes patients and controls included. The main advantage of this study was to increase the sample size and therefore the power to detect ACA clusters. However, the baseline characteristics are quite different between cases and controls in this data set in terms of age, sex, HLA genotype, and FDR status. These confounders would affect the identification of biomarkers associated with type 1 diabetes.

By contrast, the DAISY cohort includes age and sex matched controls but includes a smaller number of individuals. This study is more ideal for identifying biomarkers associated with type 1 diabetes

but is underpowered by itself for clustering ACAs. Hence, both studies were combined for ACA clustering, but only DAISY was used to identify ACA clusters associated with T1D.

## Statistical analysis

All analyses were performed using the R Studio (v2022.07.1 Build 554.pro3) on R language and environment for statistical computing (R version 4.1.2; R Foundation for Statistical Computing; www.r-project. org) and SAS (version 9.4, SAS Institute, Cary, NC). All $p$ values were two sided, a $p$ value <0.05 was considered significant. For Tables 1 and 2, continous variables were compared using Kruskal–Wallis tests; categorical variables were compared using χ2 tests. Linear regression was used to estimate mean anti-glycan antibody levels. Covariates included age, sex, HLA type, and FDR status. ROC curves were visualized using the "pROC" package. Partial r-squared was calculated from the "rsq" package. Radar/spider plots were generated using the "fmsb" package.

Our goal is to cluster ACAs into groups based on their profiles in our cohort. However, using the data as is for clustering is limited by the curse of dimensionality both in terms of the number of potential combinations of ACAs and becomes a difficult optimization problem. Thus, our first task is to identify a lower dimension graph of the data for clustering. We selected the K-nearest neighbors algorithm to construct a graph of the high dimensional data. The use of this algorithm to create a faithful topological representation is supported by extensive mathematical work described in the Uniform Manifold Approximation and Projection (UMAP) publication[65].Reiminian geometry supports the use of different distance functions for each simplex and Nerve theorem supports that the Cech simplices constructed from the K-nearest neighbors algorithm give a representation of the topology. This is advantageous to matrix factorization based approaches, like PCA, which do not preserve local data structure. The nearest neighbors and minimum distance parameters of the algorithm were selected to balance the trade off between global and local structure preservation.

Our second task is to identify clusters from this graph. While it is possible to cluster based on the UMAP embedding, the 2-dimensional embedding would lose information compared to the nearest neighbors graph since this step applies a force directed graph layout in order to give the two-dimensional representation. Cluster or community detection algorithms are often compared using different measures to determine the optimal algorithm for the specific use case. We used the louvain community detection method[66], a greedy algorithm which optimizes the modularity parameter. This method is widely used to cluster single cell sequencing data as well, where there is much sparsity, similar to our ACA dataset.

## ACA network construction

We analyzed ACAs in community clusters rather than individual models for ACAs against each glycan since many of the glycans were in functionally similar classes with overlapping structure. Therefore, ACAs against related structures would not be statistically independent from one another. Network analysis provides a stronger approach to identify associations among the ACAs.

The K-nearest neighbor (KNN) graph for ACAs was constructed for each cohort. The k-nearest neighbor algorithm was implemented from the "umap" package with $k = 5$ and the cosine metric was used as the distance measure. The graph was then constructed as an igraph object. The KNN graphs for each cohort were then merged. The edge weights unique to each graph were kept as is. For edges where both the PAGODA and DAISY graphs had weights, the product was subtracted from the summation of the weights to calculate the new weight. The Louvain clustering algorithm as implemented in the "igraph" package

was applied to identify glycan communities from the merged KNN graph.

Glycan functional classes were provided from the Consortium for Functional Glycomics and Fisher's exact test was applied to identify glycan classes enriched in the ACA clusters.

## Linear regression

Linear regression was implemented with the "lm" function in R using the model shown in Model 1.

$$Cluster\,FPC \sim Disease\,Status + Sex + First\,Degree\,Relative\,Status + HLA\,Risk + Age\,of\,Blood\,Draw \ldots\ldots\ldots\ldots \tag{1}$$

The first principal component (FPC) was calculated for each ACA cluster to assign an ACA "level" for each individual in the DAISY cohort. PCA was implemented using the "PCAtools" package. The ACA clusters were assessed for correlation along the FPC and the pairwise correlation were visualized using the "corrplot" package. The Spearman correlation was assessed and the correlation heatmap was ordered using hierarchical clustering.

## ACA discrimination models

Models to discriminate T1D status were constructed through a binomial model, implemented as a generalized linear model with a logit link function. The likelihood ratio test was applied to compare discrimination between the full (model 2) and null models (model 3).

$$Disease\,Status \sim Sex + First\,Degree\,Relative\,Status + HLA\,Risk + Age\,of\,Blood\,Draw + Cluster\,1\,FPC + Cluster\,2\,FPC \ldots\ldots + Cluster\,11\,FPC \ldots\ldots\ldots\ldots\ldots \tag{2}$$

$$Disease\,Status \sim Sex + First\,Degree\,Relative\,Status + HLA\,Risk + Age\,of\,Blood\,Draw \ldots\ldots\ldots \tag{3}$$

Random forest was implemented through the "randomForest" package in R. We calculated the q-values of the calculated AUC values using a permutation test. Specifically, we shuffled the labels of the predictor class labels and trained a full linear model (all clinical variables and the first principal component of all ACA clusters) against these randomized predictors. We repeated this for 10000 iterations and calculated the AUC for each run to establish the null distribution of AUCs in this dataset. The $q$ value was estimated as the one minus the percentile of the model AUC value against the null distribution.

## Saturation analysis

A ridge regression model as implemented in the "glmnet" package is applied with clinical variables and a set number of random glycans, with the number increasing from 1 to 200. Each number of random glycans was repeated for 1000 iterations. See the pseudocode below:

**Algorithm 1**. Saturation analysis for T1D status discrimination

 **Input**: glycan$_1$….glycan$_N$

 **Output**: AUC (T1D vs Control)

1. **Function loop** for 1, 10, 20 ….200 number of glycans:
2. Calculate AUC for binomial model (model 4) below
3. *Disease Status - Sex + First Degree Relative Status + HLA Risk + Age of Blood Draw + **Random** Glycan 1 …… +**Random** number of Glycans …………….. (4)*
4. **Repeat** 1000 times
5. **End function**

## Code reproducibility and availability

All code used for data analysis and figure generation are accessible through the github repository (https://github.com/pmtran5884/Glycancc). The code was run on both a Windows 10 and macOS Big Sur 11.6 computer with two independent users. All packages and version numbers used are provided in the supplementary information file.

## Reporting summary

Further information on research design is available in the Nature Research Reporting Summary linked to this article.

## Data availability

The processed clinical and ACA data generated in this study have been deposited on Zenodo under accession https://doi.org/10.5281/zenodo.7143430. Source data are provided with this paper.

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

## Acknowledgements

This work was supported by National Institute of Health (NIH)/ National Cancer Institute (NCI) grant 1 R21 CA199868 and U01CA221242 to J.X.S., P.G.W., R.D.C., and S.P., as well as the P41GM103694 and R24GM137763 to R.D.C. PMHT was supported by NIH/NIDDK fellowship (F30DK121461). Diabetes Autoimmunity Study in the Young (M.R., K.W., and F.D.) is supported by the National Institutes of Health (R01 DK032493 and P30 DK116073) and Helmsley Charitable Trust (G-1901-03687). The funding agencies have no role in data interpretation and publication of the results.

## Author contributions

P.M.H.T., S.P., J.X.S., P.G.W., and R.D.C. were involved with conception of the project. S.P. developed the glycan array and performed measurement of ACA in serum samples. J.X.S., R.D.C., and P.G.W. synthesized the glycan with linker. P.M.H.T., S.P., E.K., F.D., L.T., and K.P.R. were responsible for data analysis. J.X.S., D.H., K.W., M.R. contributed to clinical samples. All authors contributed to writing and editing of the paper.

## Competing interests

The authors declare no competing interests.
