## [Peer Review File · Nature Communications]

Use of a Glycomics Array to establish the Anti-carbohydrate antibody repertoire in type 1 diabetesREVIEWER COMMENTS

Reviewer #1 (Remarks to the Author):

This manuscript aims to identify biomarkers for Type 1 diabetes (T1D). The authors compared circulatory levels of anti-carbohydrate antibodies (ACAs) against 202 glycans in 278 T1D patients and 298 healthy controls. A longitudinal cohort (n=112) was also included in the study. ACA clusters enriched for aminoglycosides, blood group A and B antigens, glycolipids, ganglio-series, and O-linked glycans were determined to be associated with progression to T1D. ACAs against gentamicin, and its related structures, G418 and sisomicin, were also associated with islet autoimmunity. A key finding was that ACAs improved discrimination of T1D status of individuals over a model with only clinical variables.

Comments:

1. Although the methodology and for ACA measurement has been previously published it will be useful to include a figure that shows the reliability of the measurements. A brief description of the controls, cut-point determination for positive samples etc. should also be included.
2. Please provide a rationale for the choice of the 202 used in this study and why these provide a sufficient coverage of glycans.
3. The two cohorts PAGODA, and DAISY, are very different. You seem to recognize this and some results are obtained through the interrogation of one cohort and others by combining the two. As written however it is very difficult for the reader to distinguish between these approaches. Please add a section in Methods that clarifies which data set was used and why.
4. The discussion needs to be rewritten. Please highlight what results are novel and which confirm previous findings. Where your results diverge from previous findings (e.g., that IgG antibodies against GM2 and GT3 were not associated with islet autoimmunity or T1D) please provide a more robust discussions for the potential deviation.
5. In this study you have demonstrated that ACA can be identified in sera of patients with T1D; there is no evidence that such antibodies do indeed target insulin-secreting β -cells. The theoretical possibility of "molecular mimicry" has frequently been invoked for autoimmune diseases but has never stood up to rigorous scrutiny.

Reviewer #2 (Remarks to the Author):

In the authors manuscript, the authors describe their investigation of type 1 diabetes (T1D) blood samples from two independent cohorts of about 600 patients. They seek to demonstrate that specific anti-carbohydrate antibodies (ACAs) are more predictive for T1D progression than clinical parameters alone. The authors measured both, circulating glycans and ACAs, identified a set of 11 clusters of ACAs, and performed a glycan function enrichment analysis for each of the clusters. In this "stratification" of ACAs, five clusters have a predictive odds ratio robust enough to separate progressors vs. non-progressors of T1D. Essentially, the authors claim to have identified a set of composite biomarkers (i.e. the first principle component of the five clusters of ACA) that are predictive of T1D progression.

The full sequence of applied statistical analyses is not well described. It's hard for me to follow the decisions made, how parameters have been selected, etc. For example: Why is not directly a UMAP (and why not t-SNE or PCA?) embedding utilized followed by a n-dimensional clustering e.g. with cluster editing? Why wasn't just any clustering tool (k-means? spectral clustering? affinity propagation?) applied directly to the ACA data for cluster identification? Why not bi-clustering on glycans and ACAs? ... To be able to understand(!) the statistical data analysis decision making

procedure, a dedicated Figure is recommended. Figure 1 is too superficial.

Why was no standard machine learning approach tried? Random forests, for instance, can handle correlated features quite well in practice (or overlapping ACA structures in this particular case)... It would also allow for feature importance extraction (Gini index, for instance) directly from the model. In the presented models, many design decisions are combined, making it hard to evaluate which variables have the highest predictive power and why.

The key plot of the paper is Figure 5a, which shows the ROC plots and AUCs for the different models (clinical variables vs. ACA-cluster-first-PC-enriched composite variables). The litmus test for such models are robustness analyses. ==> How would the ROC curve / AUC look like for randomly distributed class labels (or bootstrapped labels)? I suggest to keep the input data exactly as it is, but to randomly shuffle the predictor class labels "progressor" and "non-progressor" (keep the class balance). Repeat this 5x and adds the ROC plots + mean AUC of these five plots to Fig. 5a. In a two-class problem like this one, the AUC is expected to drop to 50/50 (ROC at main diagonal). In the case of a non-heavy drop of AUC, you could also repeat the above 10000x and calculate an empirical p-value (i.e. how often is a better AUC achieved with randomized class labels? This frequency is the q-value.)

Out of interest, what would the AUC/ROC for a standard random forest be?

As this is crucial (clinical decision making deserves the highest level of reproducibility), all data and scripts need to be made available - as supplementary material including instructions on how to run the scripts to extract the composite features from the data, and how to reproduce the plots of the paper (including the scripts applied to run the robustness analysis recommended above).

The axes in all figures need labels. E.g. in Figure 2, the axes have no units (how is effect size measured? odd ratio?). The p-values are probably not ranging from 0 to 5 (log scaled)?

Reviewer #3 (Remarks to the Author):

The manuscript by Tran et al titled "The Anti-carbohydrate antibody repertoire in type 1 diabetes" is well written and posits an interesting biomarker or cluster of biomarkers to identify subjects with type 1 diabetes.

Comments

1)The statement about these analytes being stable over age comes from a small study and since DAISY has some longitudinal samples. it may have worth while checking stability using these samples.

2)The blood draw age is very different between the two cohorts and in particular between non progressors in the PAGODA, this is relevant to the question of age in 1) above

3)IgG to thymic independent antigens may be class switched by non T cell mechanisms or it may a linked T/B recognition of modified protein base by glyco or lipid moieties. Nonetheless technically I wonder about the 1:500 dilution and BSA as the matrix. Have titrations been performed and have other blockers / matrices been explored (eg FBS etc)?

4) Related to this what are the dynamic ranges, signal to noise, coefficients of variation both inter and intra- assay for this array?

5) Have the positive samples been rechecked in single plex after the multiplex array?

REVIEWER COMMENTS

Reviewer #1 (Remarks to the Author):

This manuscript aims to identify biomarkers for Type 1 diabetes (T1D). The authors compared circulatory levels of anti-carbohydrate antibodies (ACAs) against 202 glycans in 278 T1D patients and 298 healthy controls. A longitudinal cohort (n=112) was also included in the study. ACA clusters enriched for aminoglycosides, blood group A and B antigens, glycolipids, ganglio-series, and O-linked glycans were determined to be associated with progression to T1D. ACAs against gentamicin, and its related structures, G418 and sisomicin, were also associated with islet autoimmunity. A key finding was that ACAs improved discrimination of T1D status of individuals over a model with only clinical variables.

Comments:

1. Although the methodology and for ACA measurement has been previously published it will be useful to include a figure that shows the reliability of the measurements. A brief description of the controls, cut-point determination for positive samples etc. should also be included.

We appreciate this comment from the reviewer and have added Supplemental figure 1-6 to provide additional information on the methodology.

2. Please provide a rationale for the choice of the 202 used in this study and why these provide a sufficient coverage of glycans.

We have added the following statement to explain the rationale for the 202 glycans in the "Methods" section "Construction of Glycan Array" lines 326-330.

"Access to large quantities of diverse natural glycans remains a challenge in the field of glycobiology and the glycans published in ours and other similar studies represent those simple enough to be synthesized de novo or those able to be isolated from a natural source. Thus, our set of 202 glycans represents the current capability of synthetic and purification chemistry in the field of glycobiology."

3. The two cohorts PAGODA, and DAISY, are very different. You seem to recognize this and some results are obtained through the interrogation of one cohort and others by combining the two. As written however it is very difficult for the reader to distinguish between these approaches. Please add a section in Methods that clarifies which data set was used and why.

This was an oversight on our part and we thank the reviewer for this suggestion. We have added a "Analyses performed for each T1D cohort" subsection to the "Methods" section lines 362-372 to explain which dataset was used for which analysis and why.

"The samples analyzed from the PAGODA study were cross-sectional with a random selection of type 1 diabetes subjects and controls included. The main advantage of this study was to increase the sample size and therefore the power to detect ACA clusters. However, the baseline characteristics are quite different between cases and controls in this data set in terms of age, sex, HLA genotype, and FDR

status. These confounders would affect the identification of biomarkers associated with type 1 diabetes.

By contrast, the DAISY cohort includes age and sex matched controls but includes a smaller number of subjects. This study is more ideal for identifying biomarkers associated with type 1 diabetes but is underpowered by itself for clustering ACAs. Hence, both studies were combined for ACA clustering, but only DAISY was used to identify ACA clusters associated with T1D."

4. The discussion needs to be rewritten. Please highlight what results are novel and which confirm previous findings. Where your results diverge from previous findings (e.g., that IgG antibodies against GM2 and GT3 were not associated with islet autoimmunity or T1D) please provide a more robust discussions for the potential deviation.

Thank you for pointing this out. We re-wrote the section from lines 242-262 to explain more of the differences in ACAs against gangliosides.

"To our knowledge, oligosaccharides (including mono-, di-, and tri-) and blood group antigens have not been associated with progression to T1D before. Glycolipids and gangliosides have been associated with progression to T1D. However, those studies identified GM2 and GT3, whereas our study identified GD2 and GD3. The previous studies were limited by smaller sample sizes and less glycans to assess. Additionally, the previous studies used ELISA based assays whereas we applied a bead-based study. Thus, different epitopes could be exposed for these glycans. Different glycan densities can also affect antibody affinity and this can vary across assays and platforms.

While at four are gangliosides, GM2 is an a-series ganglioside, GD2 and GD3 are b-series gangliosides, GT3 is a c-series ganglioside. A-series, b-series, c-series gangliosides have 1, 2, and 3 sialic acid residues, respectively. Interestingly, a biochemical study of the glycolipids showed that GM3 makes up the majority (66.7%) of total gangliosides detected in the whole pancreas whereas an unspecified GM2 co-migrating glycolipid makes up the majority (74.2%) of total gangliosides detected in pancreatic islets. The comigrating glycolipid is likely GD2 since the number of sialic acid residues plays a strong role in glycolipid migration in both HPLC and HPTLC. This data suggests that GD2 ganglioside is expressed more in pancreatic islets and that ACAs against GD2 are associated with progression to T1D."

5. In this study you have demonstrated that ACA can be identified in sera of patients with T1D; there is no evidence that such antibodies do indeed target insulin-secreting β -cells. The theoretical possibility of "molecular mimicry" has frequently been invoked for autoimmune diseases but has never stood up to rigorous scrutiny.

This study is mainly a biomarker study to describe markers of progression to T1D. We do not claim that these ACA play a direct mechanistic role in progression to T1D. However, to address the reviewers point. We agree that the islet specific antibodies are merely biomarkers of this T cell mediated islet cell destruction. Hence, any attempt to show a causative role of identified antigens would have to show TCRs specific to that epitope. Yet, there is increasing evidence of molecular mimicry in the field of autoimmunity. Perhaps the most convincing evidence is the mimicry for EBV and multiple sclerosis (PMID: 35073561, 35025605) . Even in T1D, multiple reports have demonstrated a causal role with coxsackievirus (PMID: 219345, 17360338, 17169456, 10905477) as well as mimicry (PMID: 9498628).

Reviewer #2 (Remarks to the Author):

In the authors manuscript, the authors describe their investigation of type 1 diabetes (T1D) blood samples from two independent cohorts of about 600 patients. They seek to demonstrate that specific anti-carbohydrate antibodies (ACAs) are more predictive for T1D progression than clinical parameters alone. The authors measured both, circulating glycans and ACAs, identified a set of 11 clusters of ACAs, and performed a glycan function enrichment analysis for each of the clusters. In this "stratification" of ACAs, five clusters have a predictive odds ratio robust enough to separate progressors vs. non-progressors of T1D. Essentially, the authors claim to have identified a set of composite biomarkers (i.e. the first principle component of the five clusters of ACA) that are predictive of T1D progression.

The full sequence of applied statistical analyses is not well described. It's hard for me to follow the decisions made, how parameters have been selected, etc. For example: Why is not directly a UMAP (and why not t-SNE or PCA?) embedding utilized followed by a n-dimensional clustering e.g. with cluster editing? Why wasn't just any clustering tool (k-means? spectral clustering? affinity propagation?) applied directly to the ACA data for cluster identification? Why not bi-clustering on glycans and ACAs? ... To be able to understand(!) the statistical data analysis decision making procedure, a dedicated Figure is recommended. Figure 1 is too superficial.

We thank the reviewer for pointing this out and have provided extensive information on the theoretical foundation and the justification for our analytical choices in the "Statistical Analysis" subsection of the "Methods" section.

"Our goal is to cluster ACAs into groups based on their profiles in our cohort. However, using the data as is for clustering is limited by the curse of dimensionality both in terms of the number of potential combinations of ACAs and becomes a difficult optimization problem. Thus, our first task is to identify a lower dimension graph of the data for clustering. We selected the K-nearest neighbors algorithm to construct a graph of the high dimensional data. The use of this algorithm to create a faithful topological representation is supported by extensive mathematical work described in the UMAP publication. Reimianian geometry supports the use of different distance functions for each simplex and Nerve theorem supports that the Cech simplices constructed from the K-nearest neighbors algorithm give a representation of the topology. This is advantageous to matrix factorization based approaches like PCA, which do not preserve local data structure. The nearest neighbors and minimum distance parameters of the algorithm were selected to balance the trade off between global and local structure preservation.

Our second task is to identify clusters from this graph. While it is possible to cluster based on the UMAP embedding, the 2 dimensional embedding would lose information compared to the nearest neighbors graph since this step applies a force directed graph layout in order to give the two dimensional representation. Cluster or community detection algorithms are often compared using different measures to determine the optimal algorithm for the specific use case. The louvain community detection method is a greedy algorithm which optimizes the modularity parameter. This method is widely used to cluster single cell sequencing data as well, where there is much sparsity, similar to our ACA dataset."

Why was no standard machine learning approach tried? Random forests, for instance, can handle correlated features quite well in practice (or overlapping ACA structures in this particular case)... It would also allow for feature importance extraction (Gini index, for instance) directly from the model. In the presented models, many design decisions are

combined, making it hard to evaluate which variables have the highest predictive power and why.

A machine learning approach with each 202 variables separated would be prone to overfitting in our size data set. Our goal is not to provide the best model possible in this study with a limited sample set, but to show that ACAs provide additional information. Fitting more models to the same data would increase our odds of overfitting. We hope to use a larger data set in the future to test multiple advanced machine learning models (random forest, support vector machine, and neural nets), where we think this would be more appropriate.

The key plot of the paper is Figure 5a, which shows the ROC plots and AUCs for the different models (clinical variables vs. ACA-cluster-first-PC-enriched composite variables). The litmus test for such models are robustness analyses. ==> How would the ROC curve / AUC look like for randomly distributed class labels (or bootstrapped labels)? I suggest to keep the input data exactly as it is, but to randomly shuffle the predictor class labels "progressor" and "non-progressor" (keep the class balance). Repeat this 5x and adds the ROC plots + mean AUC of these five plots to Fig. 5a. In a two-class problem like this one, the AUC is expected to drop to 50/50 (ROC at main diagonal). In the case of a non-heavy drop of AUC, you could also repeat the above 10000x and calculate an empirical p-value (i.e. how often is a better AUC achieved with randomized class labels? This frequency is the q-value.)

We thank the reviewer for this point. We believe this adds significantly to the study and have calculated the q-values for the three models against a random distribution as suggested by the reviewer. Why have added the text below to the respective sections listed.

Methods (lines 436-441)

We calculated the q-values of the calculated AUC values using a permutation test. Specifically, we shuffled the labels of the predictor class labels and trained a full linear model (all clinical variables and the first principal component of all ACA clusters) against these randomized predictors. We repeated this for 10000 iterations and calculated the AUC for each run to establish the null distribution of AUCs in this dataset. The q-value was estimated as the 1 minus the percentile of the model AUC value against the null distribution

Results (lines 213-215)

Based on permutation testing, the full model was a robust predictor of T1D status ($q = 0.0291$) compared to the 8 ACA cluster model ($q=0.2084$) and the clinical variable only model ($q=0.9878$).

Out of interest, what would the AUC/ROC for a standard random forest be?

Please see our comment on the random forest model above

As this is crucial (clinical decision making deserves the highest level of reproducibility), all data and scripts need to be made available - as supplementary material including instructions on how to run the scripts to extract the composite features from the data, and how to reproduce the plots of the paper (including the scripts applied to run the robustness analysis recommended above).

This is all provided as a supplementary zip file.

The axes in all figures need labels. E.g. in Figure 2, the axes have no units (how is effect size measured? odd ratio?). The p-values are probably not ranging from 0 to 5 (log scaled)?

In figure 2B, The effect size is the regression coefficient, beta, in the linear model. In figure 2A, the p-values are log₁₀ transformed. These axes have been updated to provide the information the reviewer suggested.

Reviewer #3 (Remarks to the Author):

The manuscript by Tran et al titled "The Anti-carbohydrate antibody repertoire in type 1 diabetes" is well written and posits an interesting biomarker or cluster of biomarkers to identify subjects with type 1 diabetes.

Comments

1)The statement about these analytes being stable over age comes from a small study and since DAISY has some longitudinal samples. it may have worth while checking stability using these samples.

We thank the reviewer for this suggestion and have analyzed the cases with multiple samples for stability of measurement and present the data in supplemental figure 5.

"Supplemental Figure 5: Representative line plots depicting evaluation of stability of anti-carbohydrate IgG in serum samples from DAISY subjects (n=112). We selected the data from seven aminoglycosides for the analysis to represent here. Background corrected MFI (y-axis) were plotted against age of sample collection (x-axis) to show the stability of the glycan array. Serum samples (n=57) from DAISY subjects collected at multiple time points (2-4) were analyzed to evaluate the stability of the measurement of anti-carbohydrate IgG in serum."

2)The blood draw age is very different between the two cohorts and in particular between non progressors in the PAGODA, this is relevant to the question of age in 1) above

We agree with the reviewer in that these two cohorts are clinically different and so have added a new section to our methods detailing what we used each cohort for and why. We have added a "Analyses performed for each T1D cohort" subsection to the "Methods" section lines 362-372.

"The samples analyzed from the PAGODA study were cross-sectional with a random selection of type 1 diabetes subjects and controls included. The main advantage of this study was to increase the sample size and therefore the power to detect ACA clusters. However, the baseline characteristics are quite different between cases and controls in this data set in terms of age, sex, HLA genotype, and FDR status. These confounders would affect the identification of biomarkers associated with type 1 diabetes.

By contrast, the DAISY cohort includes age and sex matched controls but includes a smaller number of subjects. This study is more ideal for identifying biomarkers associated with type 1 diabetes but is underpowered by itself for clustering ACAs. Hence, both studies were combined for ACA clustering, but only DAISY was used to identify ACA clusters associated with T1D."

3)IgG to thymic independent antigens may be class switched by non T cell mechanisms or it may a linked T/B recognition of modified protein base by glyco or lipid moieties.

Nonetheless technically I wonder about the 1:500 dilution and BSA as the matrix. Have titrations been performed and have other blockers / matrices been explored (eg FBS etc)?

We have now added supplemental figures 1 and 4, which show a dilution curve for assessing the optimal dilution factor and a comparison of blocking using HSA and BSA, respectively.

“Supplementary Figure 1: Dynamic range of the glycan array determined by assaying 3 fold serum dilution starting with 100-fold dilution. Serum samples (n=5) were selected based on their highest median fluorescent intensities observed for gentamicin. Gly#96: Tobramicin, Gly#97: Amikacin, Gly#98: Gentamicin Sulfate, Gly#99: Kanamycin sulfate, Gly#100: Geneticin Disulfate salt (G418), Gly#101: Neomycin trisulfate, Gly#102: Sisomicin. Background corrected median fluorescent intensity (y-axis) is plotted against dilution factor for serum (x-axis)

Supplementary Figure 4: Comparison human serum albumin (HSA) and bovine serum albumin (BSA) as blocker for beads and for dilution of serum for analysis of anti-carbohydrate immunoglobulin-G (IgG). Selected glycans representing mono-, di, tri-saccharides and other branched glycans were evaluated for the binding of anti-carbohydrate IgG to test the blockers for conjugated beads. Conjugation of the glycans was performed as reported in the methods sections. The conjugated beads were blocked with 1% HSA (wt/v) or with 1% BSA (wt/v) in phosphate buffer containing 250mM sodium chloride (pH7.4). serum samples (n=32) were diluted 500-fold with 1% HSA and then analyzed on the blocked beads. “

4) Related to this what are the dynamic ranges, signal to noise, coefficients of variation both inter and intra- assay for this array?

The dynamic range was assessed in Supplemental Figure 1 (see point 3). The serum samples stayed linear from a 1:100 dilution to a 1:2700 dilution. We also had assessed dynamic range using lectins in serial dilution and inhibition with sugars in our original publication.

We present signal to noise ratio for select glycans in Supplemental Figure 3 and inter- and intra- assay reproducibility in Supplemental Figure 6.

“Supplementary Figure 3: Assessment of signal-to-noise ratio for analysis of anti-carbohydrate antibodies present in serum. Signal-to-noise ratio analysis was performed using serum samples (n=32) on a 45-plex array consisting of mono-, di-, tri- and other complex glycans. serum samples were diluted 500-fold with phosphate buffer pH7.4, containing 1% human serum albumin and 250mM sodium chloride.

Supplementary Figure 6: Pairs plots showing intra- (a) and inter-day (b) reproducibility of the anti aminoglycoside antibodies present in human serum (n=5). A three fold dilution curve was ran on two days for aminoglycoside beads to evaluate the reproducibility. Prior to any analysis median fluorescence intensities were log2 transformed. Intra-day reproducibility was assessed by determining coefficient of variation (CV) and correlation coefficient between two replicate wells on each day. Inter-day reproducibility was assessed by determining the CV and r values between two different days.”

5) Have the positive samples been rechecked in single plex after the multiplex array?

We have added supplemental figure 2 which compares the single plex to multiplex array.

“Supplementary Figure 2: Comparison of single-plex vs multiplex for measurement of anti-aminoglycoside antibodies in serum of human subjects (n=5). Three-fold dilution series were analyzed for comparison. Spearman’s correlation values are presented for each pairwise comparison.”

REVIEWER COMMENTS

Reviewer #1 (Remarks to the Author):

You have adequately addressed my queries.

Reviewer #2 (Remarks to the Author):

I much appreciate the amount of work that went into the robustness analysis! Very well done. All of my comments have been taking into account sufficiently and successfully - but one critical one: The machine learning part.

I do not agree with this reply:

"A machine learning approach with each 202 variables separated would be prone to overfitting in our size data set. Our goal is not to provide the best model possible in this study with a limited sample set, but to show that ACAs provide additional information. Fitting more models to the same data would increase our odds of overfitting."

Classification of 278 vs. 298 (plus the 112 longitudinal) with 202 variables in one model (e.g. a standard random forest; RF) will not lead to overfitting. RF are robust to correlated features, the number of features (m) is rather moderate compared to the number of samples (n). In transcriptomics, for instance, one often deals with 25K+ features (many highly correlated) to classify few hundreds of patients, often even <100 . Maybe my comment was not well understood. \implies I suggest to provide a RF learning package (R, or biopython, or ...) with a data matrix of the 202 features vs. the 576 (278+298) samples + the labels. How does the RF model perform? If well, what are the most predictive features (i.e. ACAs). Now take the top 10 (or top X using the Elbow method or e.g. Gini index distribution). If you now "learn" a decision tree with the best 10 (or X) ACAs: What would be its prediction performance? Is it better than using the dimensionality reduction by feature combination approach as preprocessor (as suggested in the paper)?

It is important to either explain more clearly or convincingly why such a comparison does not make sense. Or such an evaluation should be perform.

Reviewer #3 (Remarks to the Author):

My concerns have all been appropriately addressed.

Reviewer #1 (Remarks to the Author):

You have adequately addressed my queries.

>> We thank the reviewer for the time and effort.

Reviewer #2 (Remarks to the Author):

I much appreciate the amount of work that went into the robustness analysis! Very well done. All of my comments have been taken into account sufficiently and successfully - but one critical one: The machine learning part.

I do not agree with this reply:

"A machine learning approach with each 202 variables separated would be prone to overfitting in our size data set. Our goal is not to provide the best model possible in this study with a limited sample set, but to show that ACAs provide additional information. Fitting more models to the same data would increase our odds of overfitting."

Classification of 278 vs. 298 (plus the 112 longitudinal) with 202 variables in one model (e.g. a standard random forest; RF) will not lead to overfitting. RF are robust to correlated features, the number of features (m) is rather moderate compared to the number of samples (n). In transcriptomics, for instance, one often deals with 25K+ features (many highly correlated) to classify few hundreds of patients, often even <100 . Maybe my comment was not well understood. \implies I suggest to provide a RF learning package (R, or biopython, or ...) with a data matrix of the 202 features vs. the 576 (278+298) samples + the labels. How does the RF model perform? If well, what are the most predictive features (i.e. ACAs). Now take the top 10 (or top X using the Elbow method or e.g. Gini index distribution). If you now "learn" a decision tree with the best 10 (or X) ACAs: What would be its prediction performance? Is it better than using the dimensionality reduction by feature combination approach as preprocessor (as suggested in the paper)?

It is important to either explain more clearly or convincingly why such a comparison does not make sense. Or such an evaluation should be performed.

>>We thank the reviewer for his/her comments and have implemented the random forest model to our data as requested. We wanted the model to be comparable to the approach we used in our paper (PCA of clusters and linear models) and so we tested 202 ACA features with the 77 DAISY control and progressor samples. We then applied the robustness analysis that you suggested before to ensure that the random forest model is significantly different from a null distribution. We found that a random forest model of 202 ACA features to discriminate control and progressors in the DAISY cohort is not significantly different from the null distribution ($q=0.063$). We further tested the approach reviewer 2 outlined for feature selection and decision trees and found that this approach is also not significantly different from the null distribution ($q=0.214$).

We have included the code for these analyses for your review, and added a statement to our results describing these findings (pg 10-11).

CODE

```
#ACA random forest models
#Date: September 8, 2022
#Author: Paul Tran
#Updated:

rm(list = ls())

library(Glycancc)

# install.packages("caret","ranger","tidyverse","e1071")

library(caret)
library(randomForest)
library(tidyverse)
# library(e1071)

set.seed(123)

data4rf<-cbind.data.frame(daisy$igg,"Group"=daisy$pheno$Group)[-which(daisy$pheno$Group=="Non-progressor"),]
data4rf<-droplevels.data.frame(data4rf)
data4rf_rand<-data4rf
###Make null distribution for RF > top 15 predictors > classification tree pipeline###
rf_null<-c()

for(i in 1:1000){
#create data frame suitable for random forest
data4rf_rand$Group<-sample(data4rf$Group)

#run random forest as implemented in randomForest package in the caret wrapper
rf_fit <- train(as.factor(Group) ~ .,
               data = data4rf_rand,
               method = "rf",
               trControl = trainControl(method = "oob"),
               importance = TRUE)

rf_null<-c(rf_null,max(rf_fit$results$Accuracy))
print(i)
}

#save null distribution
write.csv(rf_null,"RF_null_distribution.csv")

## ACA Random Forest model #####
#run random forest as implemented in randomForest package in the caret wrapper
rf_fit <- train(as.factor(Group) ~ .,
               data = data4rf,
               method = "rf",
               trControl = trainControl(method = "oob"),
```

```
importance = TRUE)

rf_mod_acc<-max(rf_fit$results$Accuracy)

#Significance testing
qval<-function(randAUCs=rf_null,testAUC=full_roc$auc){
  qval<-1-(sum(randAUCs<as.numeric(testAUC))/1000)
  qval
}

qval(testAUC = rf_mod_acc) #0.063, rf_mod_acc = 0.662
mean(rf_null) #0.594
summary(rf_null)
sd(rf_null) #0.0387
```

Reviewer #3 (Remarks to the Author):

My concerns have all been appropriately addressed.
>> **We thank the reviewer for the time and effort.**

REVIEWERS' COMMENTS

Reviewer #2 (Remarks to the Author):

My comments have been addressed adequately. Thanks!

REVIEWERS' COMMENTS

Reviewer #2 (Remarks to the Author):

My comments have been addressed adequately. Thanks!

>> we thank the reviewers for their constructive suggestions to make this manuscript better.